# Actin-Resistant DNase1L2 as a Potential Therapeutics for CF Lung Disease

**DOI:** 10.3390/biom11030410

**Published:** 2021-03-10

**Authors:** Danila Delfino, Giulia Mori, Claudio Rivetti, Antonella Grigoletto, Gloria Bizzotto, Cristian Cavozzi, Marco Malatesta, Davide Cavazzini, Gianfranco Pasut, Riccardo Percudani

**Affiliations:** 1Department of Chemistry, Life Sciences and Environmental Sustainability, University of Parma, 43124 Parma, Italy; danila.delfino@unipr.it (D.D.); cristian.cavozzi@unipr.it (C.C.); marco.malatesta1@studenti.unipr.it (M.M.); davide.cavazzini@unipr.it (D.C.); riccardo.percudani@unipr.it (R.P.); 2Department of Pharmaceutical and Pharmacological Sciences, University of Padova, 35131 Padova, Italy; antonella.grigoletto@unipd.it (A.G.); gloria.bizzotto.2@studenti.unipd.it (G.B.)

**Keywords:** endonuclease, *Pichia pastoris*, cystic fibrosis, mucolytics, enzyme therapeutics, PEGylation

## Abstract

In cystic fibrosis (CF), the accumulation of viscous lung secretions rich in DNA and actin is a major cause of chronic inflammation and recurrent infections leading to airway obstruction. Mucolytic therapy based on recombinant human DNase1 reduces CF mucus viscosity and promotes airway clearance. However, the marked susceptibility to actin inhibition of this enzyme prompts the research of alternative treatments that could overcome this limitation. Within the human DNase repertoire, DNase1L2 is ideally suited for this purpose because it exhibits metal-dependent endonuclease activity on plasmid DNA in a broad range of pH with acidic optimum and is minimally inhibited by actin. When tested on CF artificial mucus enriched with actin, submicromolar concentrations of DNase1L2 reduces mucus viscosity by 50% in a few seconds. Inspection of superimposed model structures of DNase1 and DNase1L2 highlights differences at the actin-binding interface that justify the increased resistance of DNase1L2 toward actin inhibition. Furthermore, a PEGylated form of the enzyme with preserved enzymatic activity was obtained, showing interesting results in terms of activity. This work represents an effort toward the exploitation of natural DNase variants as promising alternatives to DNase1 for the treatment of CF lung disease.

## 1. Introduction

Chronic progressive lung disease is the dominant source of the morbidity and mortality of patients with cystic fibrosis (CF) [1,2]. As a result of the functional deficiency of the cystic fibrosis transmembrane conductance regulator (CFTR) protein, affecting salt and water transport across airway epithelia, the mucus overlying the luminal surface of the airways becomes dehydrated and abnormally viscous, thereby impairing mucociliary clearance [3,4]. The airways become vulnerable to inflammation and infection, and the combination of neutrophil influx with bacterial killing produces a massive release of DNA and filamentous actin (F-actin) that further increase the viscosity of airway secretions [5,6]. Thick and sticky mucus accumulates, and a vicious cycle begins, leading to airway obstruction, extensive lung damage, and eventually loss of pulmonary function. Therefore, secretion clearance is crucial for CF patients and could be stimulated by chest physical therapy in combination with bronchodilators and mucolytics.

Recombinant human DNase1 (rhDNase; E.C.3.1.21.1) is the most widely used mucolytic agent in people with CF [7,8]. By cleaving DNA, the enzyme reduces the abnormal viscosity of CF mucus in vitro [9] and improves airway clearance and lung function in CF patients [10,11]. Albeit poorly characterized, several endonucleases belonging to the DNase1 protein family exist in the human genome that specifically target DNA and share properties with DNase1 [12,13,14]. They all hydrolyze double-stranded DNA, yielding 5′-phosphorylated and 3′-hydroxylated oligonucleotides [15,16], and are activated by divalent cations, specifically Ca^2+^ and Mg^2+^, or Mn^2+^or Co^2+^ [17,18,19]. The variety of these DNase1-like enzymes, namely DNase1L1, DNase1L2, and DNase1L3, regulates DNase activity in different tissues and fluids according to the specific properties thereof. Indeed, at variance with DNase1 that is the most widely expressed [20], the other related DNases are typically expressed at high levels in a limited number of tissues [19,21]. Although their precise function is not yet known, the absence of any of these proteins is cause of disease, ascertaining their essential role [21]. This rich repertoire of human DNases raises the question whether other endonucleases could have a therapeutic value.

DNase1L2 is the only DNase1 family member that exerts its full activity at acidic pH [19]. Another unique feature lies in that it exists in two isoforms differing by the presence or absence of an intron coding for a proline-rich sequence that is respectively retained or removed during splicing [22]. Like DNase1, DNase1L2 has a N-terminal signal peptide predicting extracellular secretion, even though when overexpressed in human cells, only the shorter isoform was secreted, while the longer one was found in the cytoplasm [19]. DNase1L2 expression is detected at relatively low levels in many human tissues [19], but it is high in the skin and correlates with terminal differentiation of epidermal keratinocytes to corneocytes [23]. DNase1L2 knockout mice showed aberrant retention of DNA in hair, nails, and in the epithelia of the tongue and esophagus [24]. For these reasons, DNase1L2 has been associated with differentiation-associated DNA degradation in epidermis [23,24]. Furthermore, transcriptional activation of DNase1L2 by inflammatory cytokines has been demonstrated, and a potential involvement of the enzyme in DNA degradation during inflammation caused by bacterial infection has been proposed [22,25].

Interestingly, DNase1L2 was shown to be resistant to actin inhibition [19]. This is one of the major concerns with rhDNase therapy, as DNase1 depolymerizes F-actin [6] and forms a tight complex with globular actin (G-actin) which strongly inhibits endonuclease activity (Ki ∼1 nM) [26]. Although the reasons for nonresponse to rhDNase treatment in approximately 30% CF patients are still unclear [10,27,28], high actin concentrations in CF mucus could adversely affect the biological activity of rhDNase. These considerations spur the effort to find new therapeutic alternatives. Actin-resistant mutants of rhDNase, which no longer bind to G-actin, proved efficacy in reducing CF mucus viscosity in vitro [29,30]. Herein, we report the production in recombinant form of DNase1L2—a naturally occurring variant of DNase1— and provide a biochemical characterization of the endonuclease activity and actin inhibition in CF artificial mucus. In view of the successful application of PEGylation to improve the bioavailability of several therapeutic enzymes, including rhDNase [31,32,33], we applied this technique to DNase1L2. In particular, PEGylation has already demonstrated the ability to reduce the clearance of inhaled proteins and to increase their lung accumulation and residence time [34].

## 2. Materials and Methods

### 2.1. Bioinformatics

Protein sequences were downloaded from Ensembl genome browser (Release 101), aligned with CLUSTALW [35], and visualized with ESPript [36]. The frequency logo of proline-rich regions was constructed with the WebLogo server [37]. Glycosylation sites were predicted with the NetNGlyc 1.0 server [38]. Homology modeling was performed with SWISS-MODEL [39]. Analysis of the protein structure and images of atomic models were done with PyMOL (The PyMOL Molecular Graphics System, Version 1.3 Schrödinger, LLC.).

### 2.2. Recombinant Expression in Escherichia Coli

The coding sequences of DNase1L2-L (ENST00000320700.10; Q92874-1) and DNase1L2-S (ENST00000382437.8; Q92874-2) lacking the predicted signal peptide were cloned into *E. coli* expression vector pET29b (GenScript, Piscataway, NJ, USA). The constructs—pET29b-DNase1L2-L and pET29b-DNase1L2-S—were separately transformed into *E. coli* BL21 (DE3) cells by electroporation. Recombinant cells were grown in Luria-Bertani (LB) medium (10 g/L tryptone, 10 g/L NaCl, and 5 g/L yeast extract), supplemented with 50 μg/mL of kanamycin and 34 μg/mL of chloramphenicol until OD_600_ reached 0.5–0.8. Protein expression was induced with 0.5 and 1 mM isopropyl β-D-thiogalactoside (IPTG) at 20, 30, and 37 °C for 6 and 16 h. Cells were harvested by centrifugation at 4500× *g* for 15 min at 4 °C. Samples were analyzed for expression on 12% SDS-PAGE gel and visualized by Coomassie staining. Inclusion bodies enriched in recombinant protein were obtained by cells sonication, centrifugation, and solubilization in denaturing buffer with 6 M Urea, 25 mM Tris pH 7.8, and 2 mM β-mercaptoethanol (βME). Denaturing purification and on-column refolding were performed using anion-exchange chromatography. Briefly, the sample was applied to a HiTrap Q HP column (GE Healthcare) at a flow rate of 3 mL/min, followed by washing to baseline. Then, refolding of the bound protein was performed on-column by the use of a linear gradient of urea (6–0 M), followed by elution with a linear gradient of NaCl (0–1 M).

### 2.3. Integration in Pichia pastoris

The coding sequences of DNase1L2-L (ENST00000320700.10; Q92874-1) and DNase1L2-S (ENST00000382437.8; Q92874-2) lacking the predicted signal peptide were cloned into *P. pastoris* pPIC9K vector, in-frame with the α-factor signal sequence for protein secretion (GenScript, Piscataway, NJ, USA). The constructs—pPIC9K-DNase1L2-L and pPIC9K-DNase1L2-S—were transformed into *E. coli* XL1B by electroporation and then extracted and purified by alkaline lysis (see below, preparation of plasmid DNA). Each construct (20 μg) was linearized by digestion with SalI and BglII (Thermo Fisher Scientific, Waltham, MA, USA) to generate both Mut+ and MutS recombinants. The vector pPIC9K without insert was linearized and used as a negative control. Each linearized plasmid (10 μg) was separately transformed into *P. pastoris* GS115 cells by electroporation according to the instructions of the Pichia Expression Kit Manual (Invitrogen). Initially, cells were plated on histidine deficient minimal dextrose agar medium (MD; 0.4 μg/mL biotin, 1.34% yeast nitrogen base without amino acids, 2% dextrose, and 1.5% agar) and incubated at 30 °C for 2 days until His+ transformants appeared. The colonies obtained were restreaked on fresh MD plates and analyzed by PCR using 5′ and 3′ AOX1 sequencing primers (5′AOX1: 5′-GACTGGTTCCAATTGACAAGC-3′; 3′AOX1: 5′-GCAAATGGCATTCTGACATCC-3′). Briefly, one His+ colony was picked and resuspended in 25 μL of PCR buffer (1% Triton X-100, 20 mM Tris-HCl pH 8.5, and 2 mM EDTA). After the volume was adjusted to reach a final OD_600_ between 4 and 12, the sample was boiled at 95 °C for 3 min and centrifuged at 14,000 rpm for 1 min. A total of 2 μL of supernatant was used for a 50 μL PCR reaction. The majority (95%) of tested clones was Mut+ as two bands were obtained for pPIC9K-DNase1L2-L and pPIC9K-DNase1L2-S: one corresponding to the size of the DNase1L2-L and DNase1L2-S gene plus the inserted AOX1 gene in pPIC9K (854 bp + 492 bp and 791 bp + 492 bp, respectively) and the other corresponding to the AOX1gene in *P. pastoris* genome (2.2 kb). A few MutS clones (5%) were also obtained, showing only one band corresponding to the size of the DNase1L2-L and DNase1L2-S gene plus the inserted AOX1 gene in pPIC9K (854 bp + 492 bp and 791 bp + 492 bp, respectively).

### 2.4. Recombinant Expression in P. pastoris

Small-scale expression trials were used to identify optimal recombinant DNase1L2-L and DNase1L2-S expression conditions. The recombinant *P. pastoris* strains judged by PCR analysis were grown at 29 °C in 5 mL of buffered minimal glycerol medium (BMG; 1% glycerol, 100 mM potassium phosphate buffer pH 6.0, 1.34% yeast nitrogen base, and 0.4 μg/mL biotin) or buffered minimal glycerol-complex medium (BMGY; 1% yeast extract, 2% peptone, 1.34% yeast nitrogen base, 0.4 μg/mL biotin, 1% glycerol, and 100 mM potassium phosphate buffer pH 6.0) until OD600 reached 2-6 (log-phase growth). Cells were harvested by centrifugation at 3000× *g* for 6 min at room temperature, and cell pellets were resuspended to an OD_600_ of 1.0 in buffered minimal methanol medium (BMM; 0.5% methanol, 100 mM potassium phosphate buffer pH 6.0, 1.34% yeast nitrogen base, and 0.4 μg/mL biotin) or buffered methanol-complex medium (BMMY; 1% yeast extract, 2% peptone, 0.5% methanol, 100 mM potassium phosphate buffer pH 6.0, 1.34% yeast nitrogen base, and 0.4 μg/mL biotin) to induce DNase1L2 expression. The induction was conducted at different temperatures (28–30 °C) in a shaking incubator (250–300 rpm) for 24–120 h, by adding different concentrations of MeOH (0.5–1.5%) every day. Induced *P. pastoris* transformed with the empty vector was used as a control for background expression. Culture aliquots were withdrawn at every 12 or 24 h for analysis of secreted and intracellular expression. Briefly, the aliquots were centrifuged to separate culture medium (supernatant) from cells (pellet). The supernatant was concentrated by a 30 kDa centrifugal concentrator (Merck-Millipore, Burlington, MA, USA), and the pellet was lysed by boiling in 1% SDS for 10 min. Of each sample, 50 μL of concentrated supernatant and 10 μL of cell lysate were loaded onto a 12% SDS-PAGE gel and visualized by Coomassie staining.

A total of 20 colonies were tested for small-scale expression of recombinant DNase1L2-L and DNase1L2-S. The clone with the highest level of expression was selected for large-scale expression and purification.

For large-scale expression, a single colony of the DNase1L2 selected clone (pPIC9K-DNase1L2-S, His+ Mut+) was inoculated in 25 mL of BMGY and grown overnight at 30 °C in a shaking incubator (250–300 rpm). The next day, the culture was used to inoculate 1 L of fresh BMGY medium and was grown at 30 °C with vigorous shaking (300 rpm). After the culture reached an OD600 of ∼5.0, the cells were harvested, resuspended, and cultured in BMMY to a final OD600 of 1.0. MeOH was added at a final concentration of 0.5% every 24 h. After 72 h, the culture was centrifuged at 3000× *g* for 10 min at room temperature; the supernatant was collected and concentrated by a 30 kDa centrifugal concentrator (Merck-Millipore) and then stored at 4 °C until purification.

### 2.5. Protein Purification

DNase1L2 was purified by anion-exchange and size-exclusion chromatography. The concentrated supernatant was loaded, at a flow rate of 5 mL/min, onto a HiTrap Q HP column (GE Healthcare) equilibrated with 100 mM sodium phosphate buffer pH 7.4 and connected to an ÄKTA Pure FPLC System (GE Healthcare, Chicago, IL, USA). The column was washed with the same buffer and the bound protein was eluted by a two-step gradient of NaCl (0–0.2 M and 0.2–1 M). Samples were collected at various stages during purification, analyzed by 12% SDS-PAGE, and visualized by Coomassie staining. Fractions containing the purified protein were pooled and concentrated using a 30 kDa centrifugal concentrator (Merck-Millipore, Burlington, MA, USA). The concentrated protein was loaded at a flow rate of 0.5 mL/min onto a Superdex 200 10/300 HL size-exclusion column (GE Healthcare, Chicago, IL, USA), equilibrated with phosphate-buffered saline (PBS) with 1 mM CaCl_2_, and connected to an ÄKTA Pure FPLC system (GE Healthcare, Chicago, IL, USA). Protein elution was monitored by absorbance at 280 nm and by SDS-PAGE. Peak protein fractions were collected and concentrated using a 10 kDa centrifugal concentrator (Merck-Millipore, Burlington, MA, USA). The final protein concentration was estimated by absorbance at 280 nm (ε_280_ = 32890 M^−1^·cm^−1^; https://web.expasy.org/protparam/ (accessed on 31 January 2021), [40]). The purified protein was stored in PBS with 1 mM CaCl_2_ at 4 °C for at least two months or at −20 °C for several months.

### 2.6. DNA Plasmid Preparation

A plasmid DNA of approximately 3000 bp (pGEM-T) was transformed into *E. coli* XL-1 Blue cells. A positive clone was grown overnight at 37 °C in 10 mL of LB, containing 50 μg/mL of ampicillin. The culture was centrifuged at 8000× *g*, and the plasmid was isolated from the cell pellet by alkaline lysis [41,42]. Briefly, the pellet was resuspended in 1.5 mL of TE buffer (50 mM Tris-HCl pH 7.5, 10 mM EDTA) containing 100 μg/mL of RNaseA (Sigma, Kawasaki, Japan). Lysis was achieved by addition of an equal volume of 0.2 M NaOH, 1% SDS buffer. After mixing by inversion, the mixture was neutralized by addition of an equal volume of 3 M potassium acetate pH 5.5. Clarification was performed by centrifugation at 14,000 rpm for 10 min at 4 °C. A volume of the clarified lysate (supernatant) was precipitated with 0.7 volumes of isopropanol, washed with ice-cold 70% ethanol, and resuspended in DNase-free water. Re-dissolved DNA was further purified with a clean-up kit (Macherey-Nagel, Düren, Germany). DNA quality was checked by 1% agarose gel electrophoresis, and the concentration was measured by absorbance at 260 nm. Supercoiled plasmid DNA was stored at −20 °C until used for DNase activity assays.

### 2.7. DNase Activity Assays

DNase activity was measured by disappearance of supercoiled plasmid DNA in 1% agarose gel electrophoresis stained with ethidium bromide. After electrophoresis, the DNA in the gel was visualized under UV light and photographed using a ChemiDoc^®^ imager (Biorad, Hercules, CA, USA). Quantitative evaluation of DNA discrete bands was carried out by densitometric analysis with Image Lab software (Biorad, Hercules, CA, USA). DNA band intensity is inversely proportional to the amount of DNA degradation and is quantified relative to control plasmid DNA without enzyme. In this study, one unit is defined as the amount of enzyme required to completely degrade 200 ng of supercoiled plasmid DNA (3000 bp) in a total reaction volume of 10 µL in 2 min at 37 °C. This unit definition corresponds to 40 ng of our DNase1L2 and 4 ng of the purchased rhDNase (Pulmozyme; Roche, Basel, Switzerland).

A standard reaction mix containing 0.5 units of recombinant DNase1L2 (2 ng/μL; 69 nM), 20 ng/μL of pGEM-T plasmid DNA, 3 mM CaCl_2_, 3 mM MgCl_2_, and 50 mM Mes-NaOH pH 5.6 was prepared at room temperature and incubated at 37 °C for 10 min.

The effect of divalent cations on DNase activity was determined by using different concentrations (0, 1, 3, 6, and 12 mM) of MgCl_2_, CaCl_2_, MnCl_2_, and CoCl_2_.

The effect of pH was assayed under standard conditions in the 4.0–8.5 pH range, using the following buffers: 50 mM acetate-NaOH (pH 4.0 and 4.5), 50 mM MES-NaOH (pH 5.0–7.0), and 50 mM Tris-HCl (pH 7.5–8.5).

For actin inhibition assay, varying concentrations (0, 40, 80, 200, and 500 μg/mL) of G-actin from rabbit muscle (BioVision, Inc.) were preincubated for 10 min at room temperature with 0.5 units of DNase1L2 (2 ng/μL; 69 nM) in buffer containing 5 mM Tris-HCl pH 8.0, 0.4 mM ATP. Reactions were started by the addition of 20 ng/μL plasmid DNA with 3 mM CaCl2 and 3 mM MgCl_2_.

For rhDNase activity, 0.5 units of Pulmozyme (Roche; 0.2 ng/μL; 6.9 nM) were mixed with 20 ng/μL of pGEM-T plasmid DNA in 50 mM Tris-HCl pH 7, containing 3 mM CaCl_2_ and 3 mM MgCl_2_. The reaction was incubated at 37 °C for 10 min. Reactions were stopped with gel-loading buffer containing 10 mM Tris-HCl pH 7.6, 0.03% bromophenol blue, 0.03% xylene cyanol, 60% glycerol, and 60 mM EDTA. A 10-μL aliquot of each sample was analyzed by agarose gel electrophoresis.

### 2.8. Viscosity Measurements

CF artificial mucus (CF-AM) was prepared as previously described [43]. For 10 mL of CF-AM the following components were dissolved in DNase-free water: 100 mg of DNA, 50 mg of mucin, 50 mg of NaCl, 22 mg of KCl, 50 μL of egg yolk emulsion, and 200 μL of RPMI 1640 amino acids (Sigma, Kawasaki, Japan). The dispersion was equilibrated at room temperature for 2 h, and the experiments were performed within 24 h of CF-AM preparation.

Viscosity measurements were carried out with MCR302 rheometer (Anton Paar, Graz, Austria). In addition, 250 μL of CF-AM was positioned on the rheometer plate, and the upper plate was lowered to the measuring position. The temperature of the plates was controlled at 37 °C, and the shear rate was set at 1 s−1. The sample viscosity was measured for 5 or 10 min. To assay DNase activity, five units of DNase1L2 (20 ng/μL; 690 nM) or rhDNase (2 ng/μL; 69 nM) in 5 mM Tris-HCl pH 8 with 3 mM MgCl_2_ and 3 mM CaCl_2_ were added to the CF-AM sample on the rheometer plate and gently mixed. In the actin inhibition experiments, 200 μg/mL of G-actin from rabbit muscle (BioVision, Inc., Milpitas, CA, USA) was preincubated with the enzyme in 5 mM Tris-HCl pH 8 and 0.4 mM ATP for 10 min at room temperature.

### 2.9. Cysteine Reduction and Alkylation

Aliquots of recombinant DNase1L2 and rhDNase (Pulmozyme; Roche, Basel, Switzerland) were buffer exchanged into 20 mM Tris-HCl pH7.5 with 5 mM EDTA, using PD SpinTrap G-25 columns (GE Healthcare, Chicago, IL, USA) to eliminate divalent cations.

Purified DNase1L2 (14 μM) and rhDNase (14 μM) in PBS with 1 mM CaCl_2_ or in 20 mM Tris-HCl pH7.5 with 5 mM EDTA were incubated with 0 or 100 mM βME at room temperature for 30 min. After that, samples were incubated in the dark with 0 or 5 mM iodoacetamide (IAM) for 30 min. The reaction mixtures were then buffer exchanged into 20 mM Tris-HCl pH7.5, using PD SpinTrap G-25 columns (GE Healthcare, Chicago, IL, USA). Protein concentrations were determined by absorbance at 280 nm (ε_280_ = 32,890 M^−1^·cm^−1^ for DNase1L2; ε_280_ = 46,090 M^−1^·cm^−1^ for rhDNase), and DNase activity was assayed in a standard reaction mix, containing 0.5 units of enzyme, 20 ng/μL of plasmid DNA, 3 mM CaCl_2_, and 3 mM MgCl_2_.

### 2.10. Protein PEGylation

PEG24mer-NHS (Iris Biotech) was dissolved in DMSO and added twice (at time 0 and after 1 h) to purified DNase1L2 in PBS, at a final [PEG]:[protein] molar ratio of 50:1. The reaction was carried out under stirring at 25 °C for 2 h, and the synthesized conjugate was purified by size-exclusion chromatography. The reaction mixture was loaded at a flow rate of 0.5 mL/min on a Superdex 200 Increase 10/300 GL column (GE Healthcare, Chicago, IL, USA) equilibrated with PBS with 1 mM CaCl_2_ and connected to an ÄKTA Pure FPLC system (GE Healthcare). Elution was monitored by absorbance at 280 nm. PEGylated DNase1L2 was collected and concentrated with Vivaspin 15 PES (cut-off 10 kDa, Sartorius Stedim Lab.), and protein concentration was determined by absorption at 280 nm using the same extinction coefficient of the native protein (ε_280_ = 32,890 M^−1^·cm^−1^). The conjugated protein was analyzed by 12% SDS-PAGE stained with Coomassie for protein detection and with barium iodide for PEG detection. PEG24mer-DNase1L2 was assayed for activity under the same reaction conditions used for the native enzyme.

## 3. Results and Discussion

### 3.1. Expression and Purification of Recombinant DNase1L2 from Pichia Pastoris

Six transcripts are annotated by the Ensembl database for the human DNase1L2 gene (Appendix A). Three of these transcripts encode an isoform retaining a 63 bp-intron within the fifth exon (DNase1L2-L), two of them encode an isoform in which the intron is alternatively spliced (DNase1L2-S), and one transcript encodes a protein that is incomplete in N- and C-terminal ends and lacks part of the sixth exon. Expression of both DNase1L2-L and DNase1L2-S has been reported in several human tissues with the S isoform found particularly in peripheral blood leukocytes [22]. We analyzed the occurrence of the proline-rich region encoded by the 63-bp intron in vertebrates and found that it is present only in few mammalian DNase1L2 sequences and absent in sauropsids (Figure 1a and Appendix A), despite the fact that the amino acid residues are well conserved (Figure 1b).

DNase1L2 was previously produced for biochemical characterization in human cells (Shiokawa et al. 2004). Both the L and the S isoforms exhibited strong endonuclease activity in vitro on plasmid DNA with no significant differences, although a recent work reports higher activity of the S isoform [44]. Initially, we investigated the expression of recombinant DNase1L2 in a prokaryotic system. Unfortunately, DNase1L2-L and DNase1L2-S consistently formed inclusion bodies when expressed in *Escherichia coli* (Appendix A) and, although we explored different expression conditions by changing temperature, induction time, and IPTG concentration, it has not been possible to obtain either of the two isoforms in a soluble protein form. Therefore, we attempted to isolate the protein from the inclusion bodies by denaturation/renaturation procedure, but the DNase activity of the purified enzyme was negligible. Next, we decided to switch to an eukaryotic expression system such as the methylotrophic yeast *Pichia pastoris*, because it is widely used for the production of heterologous proteins for therapeutic purposes [45], and because DNase1L2 has been previously purified in this host system [25], although it is unclear which of the two isoforms was obtained. Two pPIC9K constructs were designed to encode respectively, DNase1L2-L and DNase1L2-S, fused to the α-factor sequence, thereby obtaining secretion of the protein into the culture medium (Figure 1c). Because *P. pastoris* secretes very low levels of its own proteins, the secreted heterologous protein constitutes the majority of the total proteins in the medium, thus avoiding the need to incorporate a purification tag. DNase1L2-S was efficiently expressed as a secreted protein with the expected molecular weight of approximately 28.8 kDa (Figure 1d). By contrast, in the majority of the clones tested, DNase1L2-L was not secreted (Figure 1d and Appendix A), while a weak intracellular expression was detected only in a few clones (Appendix A). Plasmid integration in the yeast genome was verified by PCR (Appendix A) and DNA sequencing. Sanger sequencing confirmed the correct sequence of the S isoform but revealed an abrupt stop in 5′ to 3′ and 3′ to 5′ reads of the L isoform wherein the retained intron lies. These outcomes led us to focus on the expression and characterization of the S isoform (hereafter referred to as DNase1L2).

Recombinant DNase1L2 was purified from the concentrated supernatant of *P. pastoris* by anion-exchange and size-exclusion chromatography. The protein eluted as a single peak from the size-exclusion chromatography column (Figure 1e). This step was necessary to eliminate contaminants, likely from the culture medium, not detected in the SDS-PAGE (Figure 1e-inset) but with an absorbance contribution at 280 nm. About 1 mg of purified protein was obtained from 1 L of culture medium. Based on our results, recombinant DNase1L2 is a nonglycosylated monomeric protein. In *P. pastoris*, recombinant proteins can be subjected to either N- or O-glycosylation [46]. The two N-glycosylation sites of human DNase1 [47] are not conserved in DNase1L2, and there are no predicted N-glycosylation sites (Appendix A). Our results indicate that the molecular weight of recombinant DNase1L2 corresponds to the value calculated from the amino acid sequence (Figure 1d,e-inset); thus, we conclude that recombinant DNase1L2 is a nonglycosylated monomeric protein.

### 3.2. Endonuclease Activity of DNase1L2 on Purified Plasmid DNA

We assayed the activity of recombinant DNase1L2 by using supercoiled plasmid DNA as substrate. Supercoiled plasmid DNA has the advantage over linear DNA to allow discrimination between single- and double-strand cuts. Like all DNase1 family enzymes, DNase1L2 has been shown to have divalent cation-dependent endonuclease activity [19]. It is long known that alkaline earth metals Ca^2+^ and Mg^2+^ have a strong synergistic effect, while transition metals Co^2+^ and Mn^2+^ are both potent activators of DNase activity [48,49]. Mg^2+^ and Ca^2+^ are typically found at concentrations of 1–3 mM and 2–4 mM, respectively, in the sputum of patients with CF [50,51], and therefore we examined the effect of these two divalent cations at these physiological concentrations. Titration of Ca^2+^ in the presence of 1 mM Mg^2+^ showed maximum activation at 1mM Ca^2+^ with a slightly inhibitory effect at higher concentrations (e.g., ~85% of activity with 3mM Ca^2+^; Figure 2a), whereas in the presence of 3 mM Mg^2+^ this inhibition was not observed (Figure 2a). This effect, reported also for DNase1 and DNase1-like enzymes [19], can be explained with the competition of Ca^2+^ and Mg^2+^ for the two Mg^2+^-binding sites located in close proximity to the active site [52]. Titration of Mg^2+^ in the presence of 1 or 3 mM Ca^2+^ showed that 3 mM Mg^2+^ was the optimal concentration (Figure 2b). Low or modest activity was detected with either Ca^2+^ or Mg^2+^ alone (<20% with 1–3 mM Ca^2+^, Figure 2a; ~30–40% with 1–3 mM Mg^2+^, Figure 2b). On the contrary, full activity was observed in the presence of either Co^2+^ or Mn^2+^ alone (Figure 2c).

DNase1L2 appeared to be unique among DNase1 family members in that it showed its maximum activity at acidic pH [19,22]. We observed that recombinant DNase1L2 was active in a wide range of pH, from 5 to 8, with optimum at pH 5.5–6 (Figure 2d). This is a key enzymatic feature for CF therapy because airway secretions from humans and pigs with CF are abnormally acidic [53,54,55,56].

Overall, our results are in accordance with previously obtained preparations of DNase1L2 from human cells [19,22]. A remarkable difference was observed in the range of pH wherein the enzyme is active. It was reported that the activity was almost negligible at pH 4–5 and pH 7–8.5, but our enzyme preparation retained substantial activity also in these pH ranges (Figure 2d).

We followed the time course of the endonuclease reaction catalyzed by DNase1L2 at the optimum pH and in the presence of the optimum concentrations of Ca^2+^ and Mg^2+^ (Figure 2e). As the supercoiled DNA rapidly disappears, circular DNA is formed first, then converted into linear DNA which progressively disappears with the formation of a smear toward the bottom of the gel. The average molecular weight of the remaining DNA cleavage products continuously decreased over time, resulting in complete DNA degradation at the end of the time course. The enzymatic behavior of DNase1L2 suggests that the enzyme operates through a single-strand nicking mechanism. The formation of the linear DNA is delayed with respect to the circular DNA because it requires the formation of two nicks on opposite DNA strands and at a short distance from each other—a few dozen base pairs. Conversely, a double-strand cut would convert the supercoiled plasmid directly into a linear DNA without the formation of circular DNA, and the linear DNA would disappear with the same rate as the supercoiled DNA. Previous investigations on the mechanism whereby DNase1 cleaves double-stranded DNA revealed that the enzyme is capable of both single- and double-cutting mode depending on the metal ion involved. Specifically, it was reported that DNase1 introduces predominantly single-strand nicks in the presence of Mg^2+^, and single- and double-strand nicks in the presence of Mn^2+^ [57] or when Ca^2+^ is added to Mg^2+^ [17,58,59]. In our experiment, the accumulation of circular DNA after supercoiled DNA degradation disclosed a single-stranded nicking mechanism, even if the slight increase of linear DNA from the beginning of the reaction, suggested the contribution, albeit minor, of double-stranded cuts that are possibly made by the enzyme.

### 3.3. DNase1L2 Reduces Viscosity of CF Artificial Mucus with Marked Resistance to Actin Inhibition

Human DNase1L2 was reported to be resistant to inhibition by actin [19], an abundant protein in neutrophils that exists in equilibrium between a monomeric (G-actin) and a polymeric form (F-actin) [60]. Polymerization/depolymerization of actin depends on several factors, such as the concentrations of G-actin, divalent cations, ATP/ADP, and the presence of specific inhibitors, one of which is the human DNase1 [61]. At the same time the binding between G-actin and DNase1 pushes the equilibrium toward depolymerization thus inhibiting DNase1 activity as well. In CF sputum, actin is commonly found in a concentration range between 0.07 and 0.2 mg/mL [32,50,51], with higher concentrations (e.g., 2 mg/mL) in some patients [6]. We analyzed the effect of actin inhibition on the activity of recombinant DNase1L2 by using plasmid DNA as substrate and actin concentrations of 0.04–0.5 mg/mL and observed only a slight reduction in activity—about 10–15%—confirming that under our experimental conditions, DNase1L2 is minimally affected by actin (Figure 3a).

To investigate the effect of DNase1L2 on mucus, we prepared CF artificial mucus (CF-AM) as previously described [43,62]. The preparation contained mucins, DNA, egg yolk, salts, and amino acids to mimic the physicochemical composition of the mucus from CF patients [51]. In particular, a high DNA concentration (10 mg/mL) was used to reproduce the viscoelastic properties of CF sputum, and amino acids were included as they are abundant in severe lung disease [63]. The viscosity of CF-AM samples was 9.8 ± 1.6 Pa·s (mean ± SD), comparable to that of CF sputa obtained from patients [32]. Five units of DNase1L2 (20 ng/μL; 690 nM) decreased the viscosity of CF-AM by ~80% after 90 s of incubation. In the presence of 0.2 mg/mL actin, DNase1L2 decreased the viscosity of CF-AM by ~50% after 90 s of incubation (Figure 3b and Appendix A). The contribution of actin to CF-AM viscosity was negligible. Next, we analyzed the activity of rhDNase under the same experimental conditions. When five units of rhDNase (2 ng/μL; 69 nM) were added to CF-AM, the viscosity was reduced by ~95% after 90 s of incubation. However, in the presence of actin (0.2 mg/mL) rhDNase reduced mucus viscosity by only ~5% after 90 s. With 50 units of rhDNase (20 ng/μL; 690 nM), the reduction was ~15% after 90 s of incubation (Figure 3b and Appendix A). These results clearly show that DNase1L2 has enhanced resistance to actin inhibition compared to rhDNase in CF artificial mucus. Indeed, with the same enzyme concentration, DNase1L2 is about four-fold more effective than rhDNase in reducing CF-AM viscosity in the presence of actin and about 10-fold more effective with the same units of activity.

### 3.4. DNase1L2 Structural Model Shows Conservation of rhDNase Active Site Residues but Significant Differences in the Actin-Binding Interface

To gain insight into the resistance of DNase1L2 to actin inhibition, we constructed a DNase1L2 structural model based on the sequence homology with human DNase1 (PDB ID: 4AWN), which was also proposed as the best template by the SWISS-MODEL server [64]. Superimposition of the DNase1L2 model (GMQE = 0.87, QMEAN4 = −0.38) with the structure of human DNase1 revealed a strong conservation of the residues involved in phosphodiester bond hydrolysis: namely, the two histidines at positions 134 and 252 with their hydrogen-bond pairs E78 and D212, the catalytic base D168, and N170, all interacting with the phosphate group [29,52] and the residues coordinating the Mg^2+^ ion in the proximity of the active site (N7, E39, D251) (Figure 4a,b). This amino acid conservation was not observed in the actin-binding region where the residues involved in the interaction appeared quite divergent (Figure 4a,c). Additional superimposition with the bovine DNase1-actin complex (PDB ID: 1ATN) allowed the inspection of the actin-DNase1/DNase1L2 interface (Figure 4c and Appendix A) that involves hydrogen bonds as well as hydrophobic and electrostatic interactions [61,65]. Modification of both polar and nonpolar interactions do occur at the actin-DNase1L2 interface, and to better understand their functional role, we also took into account the large number of DNase1 mutants that have been generated and characterized previously [29,66,67,68]. DNase1L2 β3 strand differs from the corresponding β3 strand of DNase1 by the substitution of tyrosine at position 65 with phenylalanine (F65) and by the substitution of valine at position 67 with serine (S67). Notably, V67S substitution alters the hydrophobic contact, while Y65F does not. Indeed, it has been shown that the introduction of charged or polar residues in this hydrophobic region, such as those of DNase1 mutants V67D/K and Y65R, greatly reduced actin-binding affinity, while the conservative substitutions V67M/A and Y65W did not affect binding affinity [29]. Furthermore, the substitution of the alanine at position 114 with phenylalanine (F109) introduces a steric hindrance that could interfere with actin binding. Remarkably, mutations of A114 introducing charged, aliphatic, or aromatic residues resulted in actin-resistant variants with a 10^4^-fold reduction in actin-binding affinity compared to the wild-type DNase1 [29,66]. Among the polar residues, DNase1 E13 and H44 are both substituted with aspartate in DNase1L2 (D13 and D44), while D53 is substituted by glutamate (E53) and E69 is substituted by glutamine (Q69). The substitution E13D should not significantly alter actin-binding affinity; however, it should be noted that E13 is also involved in DNA binding (Figure 4a,d [69], which is inhibited by the actin steric hindrance. Therefore, substitutions at this position are expected to alter DNase1 enzymatic activity. Indeed, the E13D mutant exhibited ~2-fold decrease in DNase activity, while E13A/K/R exhibited increased DNase activity [67]. The substitution H44D is even more noteworthy. Even though the substitution of H44 with an acidic residue did not show significant differences in actin inhibition [29], it exhibited ~20-fold reduction in DNase activity [67]. H44 is in the proximity of the negatively charged DNA phosphate, and therefore insertion of a negatively charged residue at this position could induce repulsive interactions that hinder DNA binding (Figure 4d).

In sum, the two substitutions E13D and H44D could possibly explain, at least in part, the lower enzymatic activity of DNase1L2 compared to DNase1, owing their close proximity to the DNA-binding site (Figure 4d), while the substitution A114F and, in part, also the substitution V67S could be responsible for the resistance of DNase1L2 to actin inhibition.

### 3.5. Oxidation State of DNase1L2 Conserved Cysteines

As shown above (see Figure 4a), DNase1L2 shares 54% sequence identity with DNase1. Major differences, other than the abovementioned residues involved in actin and DNA binding, include cysteine pairs that are involved in disulfide bond formation. DNase1 has two disulfide bridges, C173–C209 and C101–C104, located in close proximity of the Ca^2+^ coordination sites I and II, respectively [47,52]. Of these, the C173–C209 disulfide is deemed “essential” for the enzyme activity, owing to studies on bovine DNase1 [70,71]. DNase1L2 conserves C167 and C203, which correspond to C173 and C209 of DNase1, but lacks the entire segment—loop in the structure—comprising C101 and C104 of DNase1 (see Figure 4a). Both the deletion of the loop containing C101 and C104 and the C167 and C203 pair are conserved among DNase1L2 sequences of vertebrates (Appendix A). DNase1L2 has two other distantly located cysteines, C21 and C211, which are not present in DNase1 (see Figure 4a).

To investigate the formation of the disulfide bond between C167 and C203 in DNase1L2 and to assess its role in endonuclease activity, we performed cysteine reduction/alkylation experiments. As shown in Figure 5a, the treatment of DNase1L2 with 100 mM βME prior to incubation with divalent cations Ca^2+^ and Mg^2+^ and plasmid DNA did not significantly change DNase1L2 activity. Conversely, treatment of DNase1L2 with IAM, which alkylates free reduced cysteine thiols, drastically diminished the endonuclease activity. However, this effect could not be attributed to βME reduction, because very similar results were obtained with the native, not-reduced protein (Figure 5a). We reasoned that cysteine alkylation could interfere with the enzymatic activity by destabilizing the loop at the Ca^2+^-binding site I (see Figure 5c). Indeed, in DNase1 structure, this loop flanks the active site cleft and contains residues involved in DNA binding [52]. Consistently with our hypothesis, alkylation of both native and reduced DNase1L2 in buffer supplemented with Ca^2+^ retains the enzymatic activity almost completely (Figure 5a). These results suggest that (i) C167 and C203 are most probably reduced in DNase1L2, (ii) alkylation of these cysteine thiols prevents or alters the binding of Ca^2+^, and (iii) when Ca^2+^ is bound, the protein is protected from alkylation.

These results differ from previously published studies on the oxidation state of conserved cysteines in DNase1; however, to our knowledge, experimental evidence showing the effect of reduction and alkylation of human DNase1 has never been reported. Thus, we carried out reduction and alkylation experiments also on rhDNase. As shown in Figure 5b, treatment of rhDNase with either βME or IAM, or with βME followed by IAM, did not affect the enzyme activity, regardless of the presence of Ca^2+^. In bovine DNase1, Ca^2+^ was shown to be crucial in protecting the “essential” disulfide bond from reduction [70], and it was proposed that the cation could stabilize the conformation of the proximal loop—Ca^2+^-binding site I [72]. Since our experiments indicate that the protection of the “essential” disulfide bond in human DNase1 is not dependent on Ca^2+^, the stability of human DNase1 against reduction may arise from the amino acid composition of the loop rather than from calcium binding. Human and bovine DNase1 share six of the seven residues in the loop, whereas DNase1L2 has four different residues (Figure 5c and Appendix A). The most striking difference is found at position 206, which is occupied by proline (P206) in human DNase1, serine (S206) in bovine DNase1, and asparagine (N200) in DNase1L2. The presence of proline in the human DNase1 increases rigidity and stability of the loop that may confer protection to the “essential” disulfide bond. According to our experiments, DNase1L2 does not form stable disulfide bonds and protection of the cysteine thiols from alkylation is provided by loop stabilization upon calcium binding.

### 3.6. PEGylation Does Not Perturb DNase1L2 Endonuclease Activity

PEGylation of therapeutic proteins is an efficient approach to extend half-life and improve biological activity. In general, the attachment of PEG polymer increases the molecular mass of the protein, thereby reducing renal clearance, hides surface-exposed antigens, and shields the protein from proteolytic cleavage [31,73,74]. In the specific case of inhaled proteins, PEGylation can offer a set of useful advantages like: (i) to increase the mucus-penetrating ability, as shown for nanoparticles in CF and non-CF patients [43,75,76]; (ii) to increase protein stability in respiratory secretions that contain lung proteases [77]—PEGylation has been shown to prevent enzymatic degradation of proteins [78] and, in addition, a PEGylated version of rhDNase exhibited enhanced stability in respiratory secretions, without affecting the enzyme activity [32,33]; (iii) to increase pulmonary retention of proteins [79]; and (iv) to reduce the systemic absorption of therapy, thus being in line for a specific local therapy as in the case here reported of DNase1L2 [80].

In an initial attempt of PEGylation to evaluate the outcomes on enzymatic properties, a PEGylated form of DNase1L2 was obtained by randomly modifying the amino groups of lysines with N-hydroxysuccinimide-activated small polyethylene glycol (PEG24mer-NHS). DNase1L2 has nine surface-exposed lysines that could potentially be linked to a PEG moiety. PEG24mer-DNase1L2 conjugate was purified by size-exclusion chromatography, and PEGylation was confirmed by SDS-PAGE (Figure 6a,b). Based on chromatography retention time and electrophoretic mobility, we could estimate that an average of three out of nine lysines have been successfully modified. PEG24mer-DNase1L2 was assayed for endonuclease activity against supercoiled plasmid DNA and showed preserved or slightly increased activity compared to the native unconjugated enzyme (Figure 6c). This result is very promising in view of a possible therapeutic use of this endonuclease.

## 4. Conclusions

Mucolytic rhDNase proved effective in ameliorating airway clearance in CF patients, even though it is strongly inhibited by actin at physiological concentrations. Several strategies have been proposed to overcome actin inhibition, including the synthesis of engineered rhDNase with reduced or little affinity for actin [29] and the combination of rhDNase with gelsolin, an actin-binding protein that promotes dissociation of the actin-DNase complex [81]. The research of other human DNases that do not have such an Achilles’ heel could be a worthwhile avenue to pursue. In this work, we have chosen to investigate human DNase1L2 as a promising alternative to rhDNase for the treatment of CF lung disease. Possible advantages of this enzyme are its broad range of pH activity with acidic pH optimum and actin inhibition resistance. The acidic pH optimum is particularly desirable for an enzyme that exerts its function in the abnormally acidic CF mucus. While DNase1L2 was about 10-fold less efficient than rhDNase in plasmid digestion, when tested on CF artificial mucus in the presence of actin, it demonstrated at least a four-fold higher potency in reducing mucus viscosity compared to rhDNase. An explanation for this behavior was found by inspecting the tridimensional model of DNase1L2-actin interface wherein lie substitutions of key residues that are most probably involved in actin binding.

CF artificial mucus is a clinically relevant in vitro model capable of mimicking the properties of CF airway mucus [62]. Furthermore, its composition can be precisely controlled, and it is not influenced by interpatient variations. The use of *Pichia pastoris* as a low-cost and efficient expression system is suitable for a protein with medical purposes, although the recombinant production process of DNase1L2 is to be optimized to meet the needs of large-scale production. The clinical efficacy of protein therapeutics relies heavily on their stability and mean residence time in the body. Because protein PEGylation may favorably improve these aspects, we PEGylated DNase1L2 obtaining a preparation with a fully preserved endonuclease activity. Experiments aimed to evaluate ex vivo the mucolytic properties of both native and PEGylated DNase1L2 on mucus samples from CF patients are in progress, along with the evaluation of different PEGylation strategies.

## Figures and Tables

**Figure 1 biomolecules-11-00410-f001:**
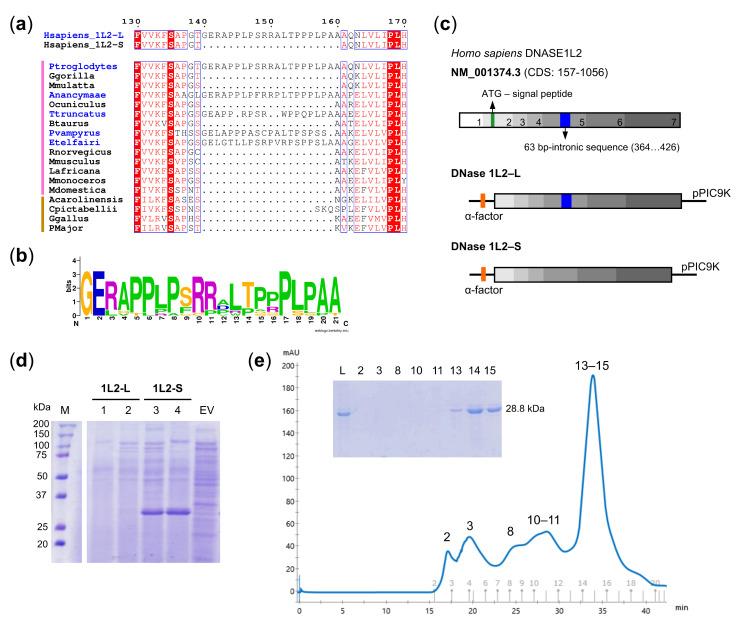
Expression and purification of DNase1L2. (**a**) Portion of multiple alignment showing the proline-rich region present only in some mammalian DNase1L2 sequences (names in blue). Vertical lines on the left indicate sequence from mammals (pink) and sauropsids (brown). (**b**) Frequency plot of the aligned proline-rich regions from thirteen mammalian DNase1L2 sequences showing remarkable conservation. (**c**) Scheme of the DNase1L2 mRNA sequence (NM_001374.3) and the constructs designed for the expression of DNase1L2-L and DNase1L2-S in *P. pastoris* GS115. CDS lacking the predicted signal peptide were cloned into pPIC9K vector bearing a sequence (α-factor) for the secretion of expressed proteins. (**d**) SDS-PAGE analysis of extracellular expression of *P. pastoris* GS115 cells transformed with pPIC9K-DNase1L2-L (two clones, 1 and 2, are shown) or with pPIC9K-DNase1L2-S (two clones, 3 and 4, are shown) or with an empty vector (EV). The band corresponding to DNase1L2-S is found at the expected molecular weight of 28.8 kDa. M: Marker. (**e**) Size-exclusion chromatogram and SDS-PAGE (inset) showing the final step of DNase1L2 purification. Numerals above peaks correspond to the gel lanes in the inset. L: Load.

**Figure 2 biomolecules-11-00410-f002:**
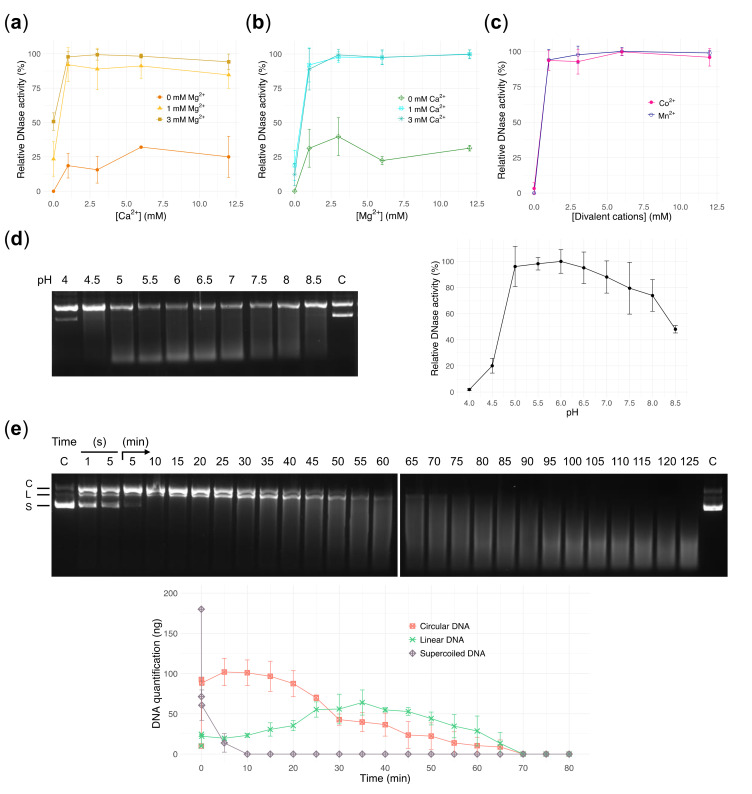
Characterization of DNase1L2 activity using purified plasmid DNA. (**a**–**c**) Gel densitometric analysis showing the effects of divalent cations on DNase1L2 activity. Data are means ± SD of three independent experiments. (**a**) Calcium titration in the presence of 0, 1, and 3 mM MgCl_2_. (**b**) Magnesium titration in the presence of 0, 1, and 3 mM CaCl_2_. (**c**) Manganese titration and cobalt titration. (**d**) Agarose gel electrophoresis (left panel) and densitometric analysis (right panel) showing the effect of pH on DNase1L2 activity. C: control, plasmid DNA without enzyme at pH 4. Data are means ± SD of three independent experiments. (**e**) Agarose gel electrophoresis (upper panel) and densitometric analysis (bottom panel) showing the time course of DNase1L2 activity. Plasmid DNA (200 ng; 90% supercoiled) was incubated with DNase1L2 (69 pmol) in the time range from 1 s to 125 min. C: control, supercoiled plasmid DNA without enzyme. Plasmid DNA forms are indicated at the left of the gel and correspond to the plots in the graph; C: circular DNA, L: linear DNA, S: supercoiled DNA. Data are means ± SD of two independent experiments.

**Figure 3 biomolecules-11-00410-f003:**
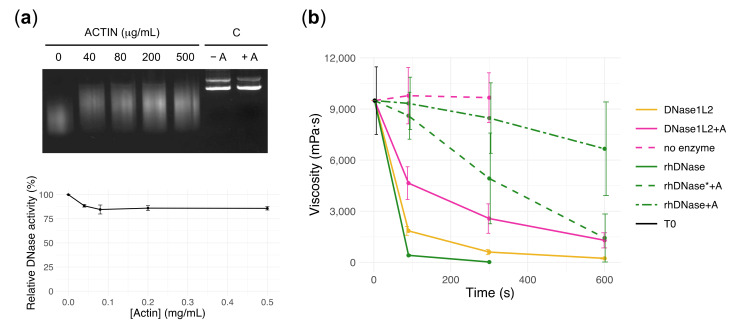
DNase1L2 reduces viscosity of cystic fibrosis (CF) artificial mucus with marked resistance to actin inhibition. (**a**) Agarose gel electrophoresis (upper panel) and densitometric analysis (bottom panel) showing the effect of actin on plasmid DNA digestion by DNase1L2. C: control, purified plasmid DNA without enzyme in the absence (-A) and in the presence of actin (+A). Data are means ± SD of two independent experiments. (**b**) Viscosity measurements showing the activity of DNase1L2 and rhDNase in CF artificial mucus (CF-AM). Activity of DNase1L2 (5 units: 20 ng/μL, pink solid and dashed line) and rhDNase (5 units: 2 ng/μL, green solid and dashed line, or 50 U: 20 ng/μL, green dashed-dot line) was tested in the presence (0.2 mg/mL) or absence of actin (A). T0 is CF-AM at the beginning of measurement. Data are means ± SD of at least two independent experiments. Viscosity values (mPa·s; mean ± SD) are reported in Appendix A.

**Figure 4 biomolecules-11-00410-f004:**
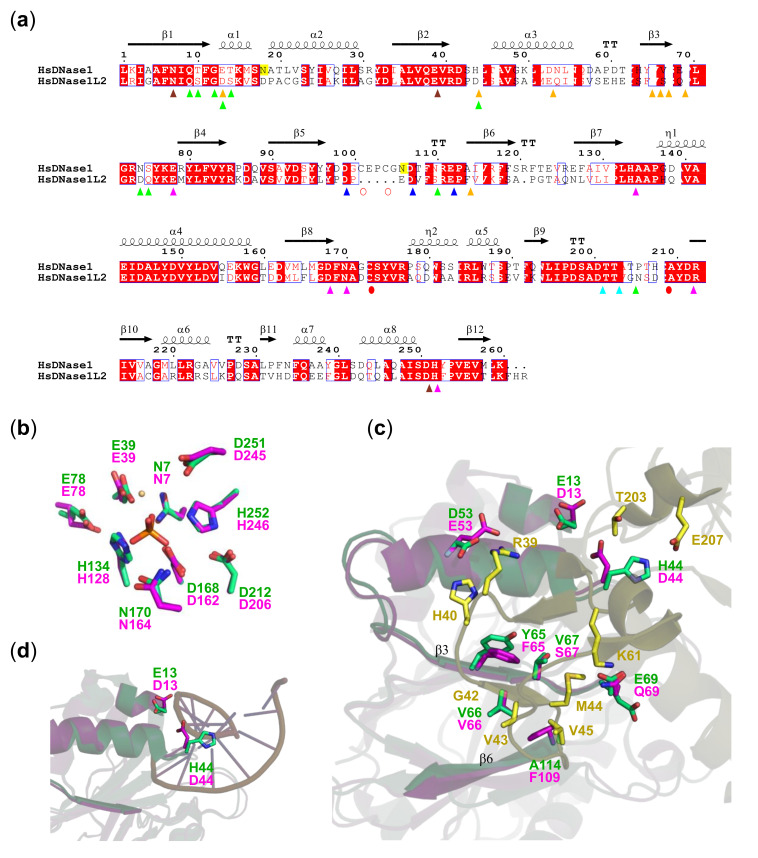
DNase1L2 exhibits conservation of DNase1 active site and diversity of the actin-binding region. (**a**) Aligned sequences of human DNase1 (HsDNase1) and DNase1L2 (HsDNase1L2). The residue numbering refers to the DNase1 sequence and is the same as in the structure in (**b**–**d**). The signal sequence that is cleaved prior to secretion in both DNases is not shown. Secondary structure elements inferred from 4AWN are shown above the alignment. Remarkable residues are indicated as follows: catalytic, pink triangles; Mg^2+^ binding, brown triangles; Ca^2+^ binding in site I, cyan triangles; Ca^2+^ binding in site II, blue triangles; actin binding, orange triangles; DNA binding, green triangles; conserved cysteines, red circles; cysteines present only in DNase1, open red circles; and N-linked glycosylation sites are highlighted in yellow. (**b**) Superimposition of the active sites of DNase1L2 homology model (pink carbons) and human DNase1 structure (4AWN; green carbons). Bound Mg^2+^ (light orange) and phosphate (orange) ions are shown. (**c**) Superimposition of the DNase1L2 homology model (pink carbons) with the human DNase1 structure (4AWN; green carbons) and the actin-bovine DNase1 complex (1ATN; yellow carbons), showing the actin-binding region. The key residues involved in interactions are shown in sticks with labels. β-strands of DNase1L2 and DNase1 interacting with actin are labeled. (**d**) Superimposition of the DNase1L2 homology model (pink carbons) with 4AWN (green carbons) and the d(GGTATACC)2-bovine DNase1 complex (1DNK; magenta carbons), showing the DNA-binding region. The key residues involved in interactions are shown in sticks with labels.

**Figure 5 biomolecules-11-00410-f005:**
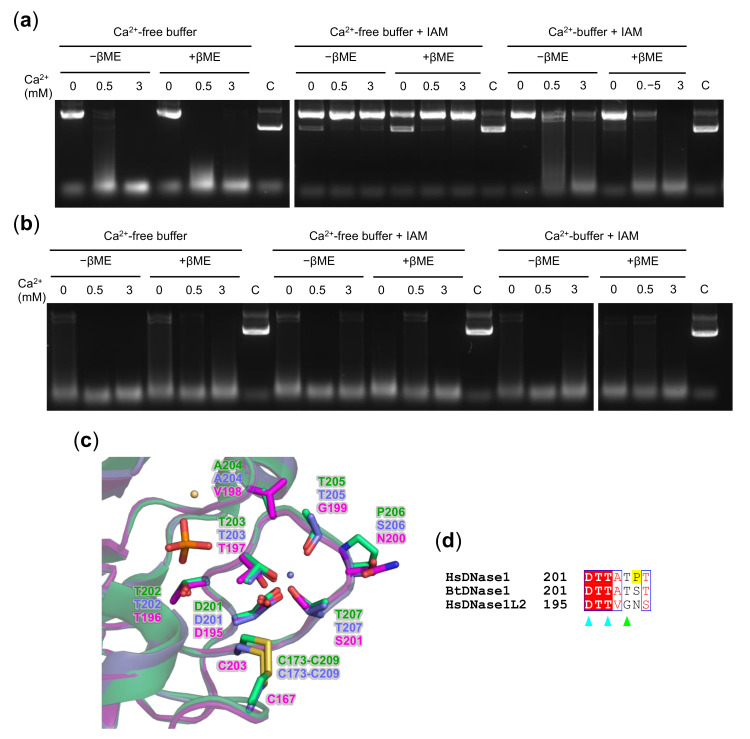
Ca^2+^ protection of DNase1L2 reduced cysteines from alkylation. Agarose gel electrophoresis showing the effect of β–mercaptoethanol (βME) and iodoacetamide (IAM) on plasmid DNA digestion by (**a**) DNase1L2 and (**b**) rhDNase. The enzymes were in buffer supplemented with 1 mM CaCl_2_ or in CaCl_2_-free buffer. After treatment, reactions were assembled as described in Materials and Methods. (**c**) Superimposition of the DNase1L2 model (pink carbons) with the human DNase1 structure (4AWN; green carbons) and the bovine DNase1 structure (1ATN; violet carbons), showing the calcium-binding site I. Bound Ca^2+^ (violet), Mg^2+^ (light orange), and phosphate (orange) ions are shown. (**d**) Aligned sequences of human DNase1 (HsDNase1), bovine DNase1 (BtDNase1), and human DNase1L2 (HsDNase1L2). The position of the first amino acid of each sequence is on the right. The residue numbering is the same as in the structure in (**c**). Key residues are indicated as follows: Ca^2+^-binding site I, cyan triangles; DNA binding, green triangle; and the proline (P206) discussed in the text is highlighted in yellow.

**Figure 6 biomolecules-11-00410-f006:**
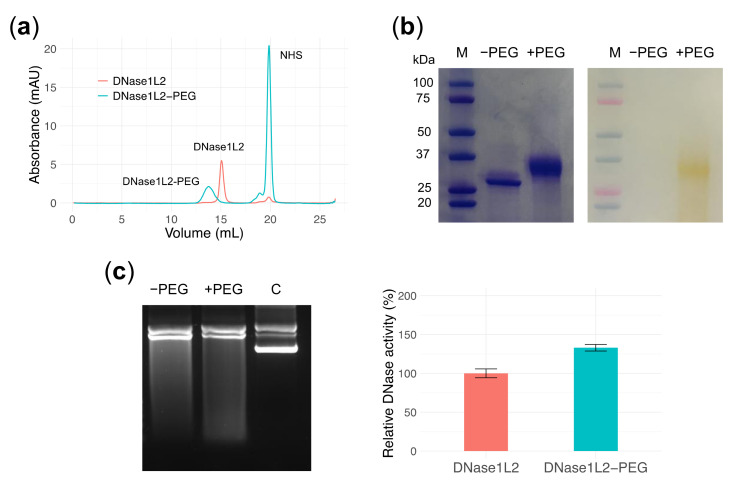
PEGylation of DNase1L2. (**a**) Size-exclusion chromatogram of DNase1L2 (salmon) and reaction mixture of DNase1L2 and PEG-NHS (cyan). Representative peaks are labelled. (**b**) SDS-PAGE of native and PEGylated DNase1L2 with Coomassie (left panel) and PEG-specific barium iodide dye (right panel) staining. M: marker. (**c**) Agarose gel electrophoresis (left panel) and densitometric analysis (right panel) showing DNase1L2 and DNase1L2-PEG activity on supercoiled plasmid DNA. Data are means ± SD of two independent experiments.

## Data Availability

Raw data generated in this study are deposited in the Harvard dataverse repository (https://doi.org/10.7910/DVN/PUVRCV (accessed on 31 January 2021)).

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
