# Peer review of "Actin-Resistant DNase1L2 as a Potential Therapeutics for CF Lung Disease"

_biomolecules, 2021, doi:10.3390/biom11030410_

Round 1

Reviewer 1 Report

I found this to be a very entertaining and informative submission which was quite carefully constructed.  I have only one major comment, and that is with respect to the justification/discussion of pegylation of DNase1L2.  The justification for pegylation (that it increases half life) is probably not relevant to topical administration to the airways, and the observation that pegylation of a small molecule (ibuprofen) increases mucus penetration is also probably irrelevant with respect to predicting the effect of pegylation on a recombinant protein. What IS important, however, is how this unglycosylated recombinant product from Pichia pastoris behaves when aerosolized, and whether these properties are affected by pegylation, something that the authors have failed to consider. 

Minor comments:

Line 46: should read "people" with CF

Line 308: should eaither read "despite the fact" or "in spite of the fact"

Lines 317-318: I assume this should read "it has notbeen possible to obtain either of the two isomforms..."

Author Response

Point 1: I found this to be a very entertaining and informative submission which was quite carefully constructed.  I have only one major comment, and that is with respect to the justification/discussion of pegylation of DNase1L2.  The justification for pegylation (that it increases half life) is probably not relevant to topical administration to the airways, and the observation that pegylation of a small molecule (ibuprofen) increases mucus penetration is also probably irrelevant with respect to predicting the effect of pegylation on a recombinant protein. What IS important, however, is how this unglycosylated recombinant product from Pichia pastoris behaves when aerosolized, and whether these properties are affected by pegylation, something that the authors have failed to consider. 

Response 1: We thank the referee for the review of the manuscript and for valuable suggestions.

We want firstly to specify that the cited reference is not related to a PEGylation of a small molecule, ibuprofen, but is instead reporting the PEGylation of nanoparticles and how this approach allowed mucus penetration of these nanoparticles loaded with ibuprofen.

To better highlight the advantages of PEGylation and the rational of its use in this context we implemented both the “Introduction” and the “Results and Discussion” sections as follows:

  • At the end of the “Introduction” we added the following underlined sentence “In view of the successful application of PEGylation to improve the bioavailability of several therapeutic enzymes, including rhDNase[31–33], we applied this technique to DNase1L2. In particular, PEGylation has already demonstrated the ability to reduce the clearance of inhaled proteins and to increase their lung accumulation and residence time [Koussoroplis SJ, Paulissen G, Tyteca D, Goldansaz H, Todoroff J, Barilly C, et al. PEGylation of antibody fragments greatly increases their local residence time following delivery to the respiratory tract. J Control Release. 2014;187:91–100.].”
  • In the “Results and Discussion” section, subsection 3.6 PEGylation Does Not Perturb DNase1L2 Endonuclease Activity, we added the following underlined text: “PEGylation of therapeutic proteins is an efficient approach to extend half-life and improve biological activity. In general, the attachment of PEG polymer increases the molecular mass of the protein, thereby reducing renal clearance, hides surface-exposed antigens and shields the protein from proteolytic cleavage[31,73,74]. In the specific case of inhaled proteins, PEGylation can offer a set of useful advantages like: i) to increase the mucus-penetrating ability, as shown for nanoparticles in CF and non-CF patients [43, Wang Y-Y, Lai SK, Suk JS, Pace A, Cone R, Hanes J. Addressing the PEG mucoadhesivity paradox to engineer nanoparticles that “slip” through the human mucus barrier. Angew Chem Int Ed Engl. 2008;47(50):9726–9.; Kim AJ, Boylan NJ, Suk JS, Hwangbo M, Yu T, Schuster BS, et al. Use of single-site-functionalized PEG dendrons to prepare gene vectors that penetrate human mucus barriers. Angew Chem Int Ed Engl. 2013;52(14):3985–8.], ii) to increase protein stability in respiratory secretions that contain lung proteases [Chakraborti S, Sarkar J, Pramanik PK, Chakraborti T. Role of proteases in lung disease: a brief overview. In: Proteases in human diseases. 2017:333–74.]—PEGylation has been shown to prevent enzymatic degradation of proteins [Zhang C, Desai R, Perez-Luna V, Karuri N. PEGylation of lysine residues improves the proteolytic stability of fibronectin while retaining biological activity. Biotechnol J. 2014;9(8):1033–43.] and, in addition, a PEGylated version of rhDNase exhibited enhanced stability without affecting the enzyme activity[32,33], iii) to increase pulmonary retention of proteins [Freches D, Patil HP, Franco MM, Uyttenhove C, Heywood S, Vanbever R. PEGylation prolongs the pulmonary retention of an anti-IL-17A fab’ antibody fragment after pulmonary delivery in three different species. Int J Pharm. 2017;521(1–2):120–9.], and iv) to reduce the systemic absorption of therapy, thus being in line for a specific local therapy as in the case here reported of DNase1L2 [Ralph W. Niven, K. Lane Whitcomb, Linda Shaner, Anna Y. Ip & Olaf B. Kinstler . The Pulmonary Absorption of Aerosolized and Intratracheally Instilled rhG-CSF and monoPEGylated rhG-CSF. Pharmaceutical Research volume 12, pages1343–1349(1995)].

Minor comments:

Line 46: should read "people" with CF

Line 46 has been corrected as suggested, "people" with CF

Line 308: should either read "despite the fact" or "in spite of the fact"

Line 308 has been corrected with "despite the fact"

Lines 317-318: I assume this should read "it has not been possible to obtain either of the two isoforms..."

Line 317-318 has been corrected as suggested, "it has not been possible to obtain either of the two isoforms..."

Reviewer 2 Report

This is a very well written and executed work. 

One major interesting issue I was thinking of was the CF-artificial mucus. Could the authors discuss or present the relevance of the content in the CF-AM? How relevant is this to original human CF mucus? What are the boundaries and possibilities? 

The abberviatio of B-mercaptoethanol is explained several times and should only appear once. Continue to use the ßME further on. 

Very detailed M&Methods description. I assume this is neccessery if other researchers aim to reproduce the data. 

Solid points to bring up in the disucussion when linking the low pH present in the CF human mucus and this works testing the pH relevance.   

Author Response

Point 1: This is a very well written and executed work. 

One major interesting issue I was thinking of was the CF-artificial mucus. Could the authors discuss or present the relevance of the content in the CF-AM? How relevant is this to original human CF mucus? What are the boundaries and possibilities? 

Response 1: We thank the referee for the review of the manuscript and for valuable suggestions.

The content and the clinical relevance of the CF-AM have been specified in the revised version. In the “Results and Discussion” section, subsection 3.3. DNase1L2 Reduces Viscosity of CF Artificial Mucus with Marked Resistance to Actin Inhibition, we added the following underlined sentence: “To investigate the effect of DNase1L2 on mucus, we prepared CF artificial mucus (CF-AM) as previously described [43, Yang Y, Tsifansky MD, Wu CJ, Yang HI, Schmidt G, Yeo Y. Inhalable antibiotic delivery using a dry powder co-delivering recombinant deoxyribonuclease and ciprofloxacin for treatment of cystic fibrosis. Pharm Res. 2010 Jan;27(1):151-60]. The preparation contained mucins, DNA, egg yolk, salts, and amino acids to mimic the physicochemical composition of the mucus from CF patients[51]. In particular, a high DNA concentration (10 mg/mL) was used to reproduce the viscoelastic properties of CF sputum, and amino acids were included as they are abundant in severe lung disease [Thomas SR, Ray A, Hodson ME, Pitt TL. Increased sputum amino acid concentrations and auxotrophy of Pseudomonas aeruginosa in severe cystic fibrosis lung disease. Thorax. 2000 Sep;55(9):795-7].” In the “Conclusion” section we added the following comment: “CF artificial mucus is a clinically relevant in vitro model capable of mimicking the properties of CF airway mucus [Yang Y, Tsifansky MD, Wu CJ, Yang HI, Schmidt G, Yeo Y. Inhalable antibiotic delivery using a dry powder co-delivering recombinant deoxyribonuclease and ciprofloxacin for treatment of cystic fibrosis. Pharm Res. 2010 Jan;27(1):151-60]. Furthermore, its composition can be precisely controlled and it is not influenced by inter-patient variations.”

The citation that we added refers to a work in which the CF artificial mucus has been used as a model to test the co-delivery of antibiotics and rhDNase [Yang Y, Tsifansky MD, Wu CJ, Yang HI, Schmidt G, Yeo Y. Inhalable antibiotic delivery using a dry powder co-delivering recombinant deoxyribonuclease and ciprofloxacin for treatment of cystic fibrosis. Pharm Res. 2010 Jan;27(1):151-60].

Point 2: The abbreviation of B-mercaptoethanol is explained several times and should only appear once. Continue to use the ßME further on. 

Response 2: Redundant explanations have been substituted by the ßME abbreviation.

Point 3: Very detailed M&Methods description. I assume this is necessary if other researchers aim to reproduce the data. 

Response 3: We agree with the reviewer that a detailed description of M&Methods can help other researchers to reproduce the data.

Point 4: Solid points to bring up in the discussion when linking the low pH present in the CF human mucus and this works testing the pH relevance.   

Response 4: We thank the reviewer for this comment. We added this point in the “Conclusion” section:

“The acidic pH optimum is particularly desirable for an enzyme that exerts its function in the abnormally acidic CF mucus.”